# Spectral Temporal Graph Neural Network for Multivariate Time-series Forecasting

**Defu Cao**[1,*,†], **Yujing Wang**[1,2,†], **Juanyong Duan**[2], **Ce Zhang**[3], **Xia Zhu**[2]
**Conguri Huang**[2], **Yunhai Tong**[1], **Bixiong Xu**[2], **Jing Bai**[2], **Jie Tong**[2], **Qi Zhang**[2]
[1]Peking University   [2]Microsoft   [3]ETH Zürich
{cdf, yujwang, yhtong}@pku.edu.cn   ce.zhang@inf.ethz.ch
{juaduan, zhuxia, conhua, bix, jbai, jietong, qizhang}@microsoft.com

## Abstract

Multivariate time-series forecasting plays a crucial role in many real-world applications. It is a challenging problem as one needs to consider both intra-series temporal correlations and inter-series correlations simultaneously. Recently, there have been multiple works trying to capture both correlations, but most, if not all of them only capture temporal correlations in the time domain and resort to pre-defined priors as inter-series relationships.

In this paper, we propose Spectral Temporal Graph Neural Network (StemGNN) to further improve the accuracy of multivariate time-series forecasting. StemGNN captures inter-series correlations and temporal dependencies *jointly* in the *spectral domain*. It combines Graph Fourier Transform (GFT) which models inter-series correlations and Discrete Fourier Transform (DFT) which models temporal dependencies in an end-to-end framework. After passing through GFT and DFT, the spectral representations hold clear patterns and can be predicted effectively by convolution and sequential learning modules. Moreover, StemGNN learns inter-series correlations automatically from the data without using pre-defined priors. We conduct extensive experiments on ten real-world datasets to demonstrate the effectiveness of StemGNN.

## 1 Introduction

Time-series forecasting plays a crucial role in various real-world scenarios, such as traffic forecasting, supply chain management and financial investment. It helps people to make important decisions if the future evolution of events or metrics can be estimated accurately. For example, we can modify our driving route or reschedule an appointment if there is a severe traffic jam anticipated in advance. Moreover, if we can forecast the trend of COVID-19 in advance, we are able to reschedule important events and take quick actions to prevent the spread of epidemic.

Making accurate forecasting based on historical time-series data is challenging, as both intra-series temporal patterns and inter-series correlations need to be modeled *jointly*. Recently, deep learning models shed new lights on this problem. On one hand, Long Short-Term Memory (LSTM) [10], Gated Recurrent Units (GRU) [6], Gated Linear Units (GLU) [7] and Temporal Convolution Networks (TCN) [3] have achieved promising results in temporal modeling. At the same time, Discrete Fourier Transform (DFT) is also useful for time-series analysis. For instance, State Frequency Memory (SFM) network [32] combines the advantages of DFT and LSTM jointly for stock price prediction; Spectral Residual (SR) model [23] leverages DFT and achieves state-of-the-art performances in

---

[*]The work was done when the author did internship at Microsoft.
[†]Equal Contribution

time-series anomaly detection. Another important aspect of multivariate time-series forecasting is to model the correlations among multiple time-series. For example, in the traffic forecasting task, adjacent roads naturally interplay with each other. Current state-of-the-art models highly depend on Graph Convoluational Networks (GCNs) [13] originated from the theory of Graph Fourier Transform (GFT). These models [31, 17] stack GCN and temporal modules (e.g., LSTM, GRU) directly, which only capture temporal patterns in the time domain and require a pre-defined topology of inter-series relationships.

In this paper, our goal is to better model the intra-series temporal patterns and inter-series correlations jointly. Specifically, we hope to combine *both* the advantages of GFT and DFT, and model multivariate time-series data entirely in the *spectral domain*. The intuition is that after GFT and DFT, the spectral representations could hold clearer patterns and can be predicted more effectively. It is non-trivial to achieve this goal. The key technical contribution of this work is a carefully designed StemGNN (Spectral Temporal Graph Neural Network) block. Inside a StemGNN block, GFT is first applied to transfer structural multivariate inputs into spectral time-series representations, while different trends can be decomposed to *orthogonal* time-series. Furthermore, DFT is utilized to transfer each univariate time-series into the frequency domain. After GFT and DFT, the spectral representations become easier to be recognized by convolution and sequential modeling layers. Moreover, a latent correlation layer is incorporated in the end-to-end framework to learn inter-series correlations *automatically*, so it does not require multivariate dependencies as priors. Moreover, we adopt both forecasting and backcasting output modules with a shared encoder to facilitate the representation capability of multivariate time-series.

The **main contributions** of this paper are summarized as follows:

- To the best of our knowledge, StemGNN is the first work that represents both intra-series and inter-series correlations jointly in the *spectral domain*. It encapsulates the benefits of GFT, DFT and deep neural networks simultaneously and collaboratively. Ablation studies further prove the effectiveness of this design.

- StemGNN enables a data-driven construction of dependency graphs for different time-series. Thereby the model is general for all multivariate time-series without pre-defined topologies. As shown in the experiments, automatically learned graph structures have good interpretability and work even better than the graph structures defined by humans.

- StemGNN achieves state-of-the-art performances on nine public benchmarks of multivariate time-series forecasting. On average, it outperforms the best baseline by 8.1% on MAE an 13.3% on RMSE. A case study on COVID-19 further shows its feasibility in real scenarios.

## 2   Related Work

Time-series forecasting is an emerging topic in machine learning, which can be divided into two major categories: *univariate techniques* [20, 22, 18, 27, 32, 19, 18] and *multivariate techniques* [24, 21, 17, 31, 3, 29, 25, 16, 15]. *Univariate techniques* analyze each individual time-series separately without considering the correlations between different time-series [22]. For example, FC-LSTM [30] forecasts univariate time-series with LSTM and fully-connected layers. SMF [32] improves the LSTM model by breaking down the cell states of a given univariate time-series into a series of different frequency components through Discrete Fourier Transform (DFT). N-BEATS [19] proposes a deep neural architecture based on a deep stack of fully-connected layers with basis expansion.

*Multivariate techniques* consider a collection of multiple time-series as a unified entity [24, 9]. TCN [3] is a representative work in this category, which treats the high-dimensional data entirely as a tensor input and considers a large receptive field through dilated convolutions. LSTNet [14] uses convolution neural network (CNN) and recurrent neural network (RNN) to extract short-term local dependence patterns among variables and discover long-term patterns of time series. DeepState [21] marries state space models with deep recurrent neural networks and learns the parameters of the entire network through maximum log likelihood. DeepGLO [25] leverages both global and local features during training and forecasting. The global component in DeepGLO is based on matrix factorization and is able to capture global patterns by representing each time-series as a linear combination of basis components. There is another category of works using graph neural networks to capture the correlations between different time-series explicitly. For instance, DCRNN [17]

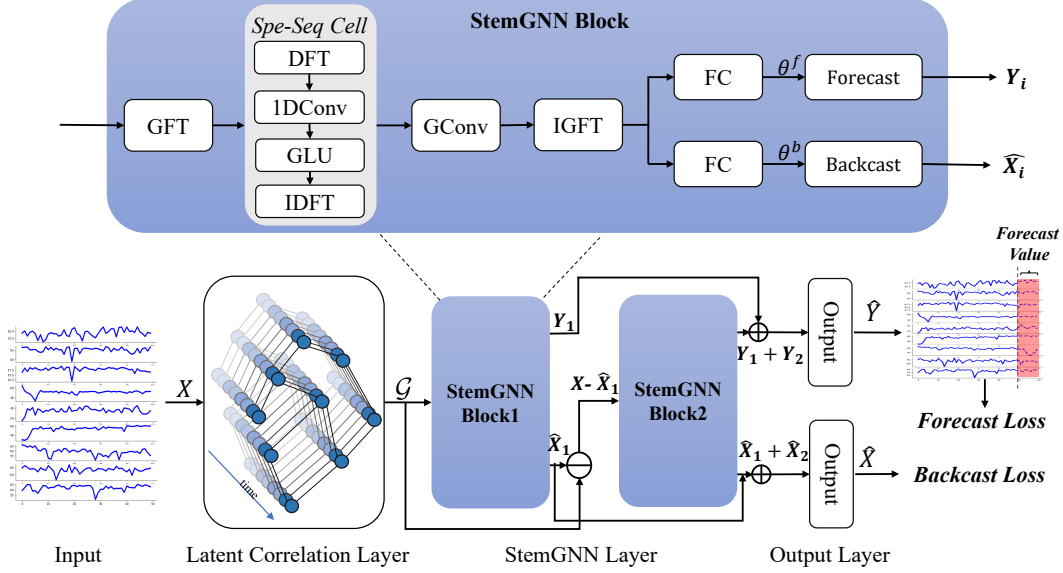

Figure 1: The overall architecture of Spectral Temporal Graph Neural Network.

incorporates both spatial and temporal dependencies in the convolutional recurrent neural network for traffic forecasting. ST-GCN [31] is another deep learning framework for traffic prediction, integrating graph convolution and gated temporal convolution through spatio-temporal convolutional blocks. GraphWaveNet [29] combines graph convolutional layers with adaptive adjacency matrices and dilated casual convolutions to capture spatio-temporal dependencies. However, most of them either ignore the inter-series correlations or require a dependency graph as priors. In addition, although Fourier transform has showed its advantages in previous works, none of existing solutions capture temporal patterns and multivariate dependencies jointly in the *spectral domain*. In this paper, StemGNN is proposed to address these issues. We refer you to recent surveys [28, 34, 33] for more details about related works.

## 3 Problem Definition

In order to emphasize the relationships among multiple time-series, we formulate the problem of multivariate time-series forecasting based on a data structure called *multivariate temporal graph*, which can be denoted as $\mathcal{G} = (X, W)$. $X = \{x_{it}\} \in \mathbb{R}^{N \times T}$ stands for the multivariate time-series input, where $N$ is the number of time-series (nodes), and $T$ is the number of timestamps. We denote the observed values at timestamp $t$ as $X_t \in \mathbb{R}^N$. $W \in \mathbb{R}^{N \times N}$ is the adjacency matrix, where $w_{ij} > 0$ indicates that there is an edge connecting nodes $i$ and $j$, and $w_{ij}$ indicates the strength of this edge.

Given observed values of previous $K$ timestamps $X_{t-K}, \cdots, X_{t-1}$, the task of *multivariate time-series forecasting* aims to predict the node values in a multivariate temporal graph $\mathcal{G} = (X, W)$ for the next $H$ timestamps, denoted by $\hat{X}_t, \hat{X}_{t+1}, \cdots, \hat{X}_{t+H-1}$. These values can be inferred by the forecasting model $F$ with parameter $\Phi$ and a graph structure $\mathcal{G}$, where $\mathcal{G}$ can be input as prior or automatically inferred from data.

$$\hat{X}_t, \hat{X}_{t+1}..., \hat{X}_{t+H-1} = F(X_{t-K}, ..., X_{t-1}; \mathcal{G}; \Phi). \tag{1}$$

## 4 Spectral Temporal Graph Neural Network

### 4.1 Overview

Here, we propose Spectral Temporal Graph Neural Network (StemGNN) as a general solution for multivariate time-series forecasting. The overall architecture of StemGNN is illustrated in Figure

1. The multivariate time-series input $X$ is first fed into a latent correlation layer, where the graph structure and its associated weight matrix $W$ can be inferred automatically from data.

Next, the graph $\mathcal{G} = (X, W)$ serves as input to the StemGNN layer consisting of two residual StemGNN blocks. A StemGNN block is by design to model structural and temporal dependencies inside multivariate time-series jointly in the spectral domain (as visualized in the top diagram of Figure 1). It contains a sequence of operators in a well-designed order. First, a Graph Fourier Transform (GFT) operator transforms the graph $\mathcal{G}$ into a spectral matrix representation, where the univariate time-series for each node becomes linearly independent. Then, a Discrete Fourier Transform (DFT) operator transforms each univariate time-series component into the frequency domain. In the frequency domain, the representation is fed into 1D convolution and GLU sub-layers to capture feature patterns before transformed back to the time domain through inverse DFT. Finally, we apply graph convolution on the spectral matrix representation and perform inverse GFT.

After the StemGNN layer, we add an output layer composed of GLU and fully-connected (FC) sub-layers. There are two kinds of outputs in the network. The forecasting outputs $Y_i$ are trained to generate the best estimation of future values, while the backcasting outputs $\hat{X}_i$ are used in an auto-encoding fashion to enhance the representation power of multivariate time-series. The final loss function can be formulated as a combination of both forecasting and backcasting losses:

$$\mathcal{L}(\hat{X}, X; \Delta_\theta) = \sum_{t=0}^{T} ||\hat{X}_t - X_t||_2^2 + \sum_{t=K}^{T} \sum_{i=1}^{K} ||B_{t-i}(X) - X_{t-i}||_2^2 \qquad (2)$$

where the first term represents for the forecasting loss and the second term denotes the backcasting loss. For each timestamp $t$, $\{X_{t-K}, ..., X_{t-1}\}$ are input values within a sliding window, and $X_t$ is the ground truth value to forecast; $\hat{X}_t$ is the forecasted value for the timestamp $t$, and $\{B_{t-K}(X), ..., B_{t-1}(X)\}$ are reconstructed values from the backcasting module. $B$ indicates the entire network that generates backcasting output, $\Delta_\theta$ denotes all parameters in the network.

In the inference phase, we adopt a rolling strategy for multi-step prediction. First, $\hat{X}_t$ is predicted by taking $\{X_{t-K}, ..., X_{t-1}\}$ as input. Then, the input will be changed to $\{X_{t-K+1}, ..., X_{t-1}, \hat{X}_t\}$ for predicting the next timestamp $\hat{X}_{t+1}$. By applying this rolling strategy consecutively, we can obtain forecasting values of the next $H$ timestamps.

### 4.2 Latent Correlation Layer

GNN-based approach requires a graph structure when modeling multivariate time-series. It can be constructed by human knowledge (such as road network in traffic forecasting), but sometimes we do not have a pre-defined graph structure as prior. In order to serve general cases, we leverage the self-attention mechanism to learn latent correlations between multiple time-series automatically. In this way, the model emphasizes task-specific correlations in a data-driven fashion.

First, the input $X \in \mathbb{R}^{N \times T}$ is fed into a Gated Recurrent Unit (GRU) layer, which calculates the hidden state corresponding to each timestamp $t$ sequentially. Then, we use the last hidden state $R$ as the representation of entire time-series and calculate the weight matrix $W$ by the self-attention mechanism as follows,

$$Q = RW^Q, K = RW^K, W = \text{Softmax}(\frac{QK^T}{\sqrt{d}}) \qquad (3)$$

where $Q$ and $K$ denote the representation for query and key, which can be calculated by linear projections with learnable parameters $W^Q$ and $W^K$ in the attention mechanism, respectively; and $d$ is the hidden dimension size of $Q$ and $K$. The output matrix $W \in \mathbb{R}^{N \times N}$ is served as the adjacency weight matrix for graph $\mathcal{G}$. The overall time complexity of self-attention is $O(N^2 d)$.

### 4.3 StemGNN Block

The StemGNN layer is constructed by stacking multiple *StemGNN blocks* with skip connections. A *StemGNN block* is designed by embedding a Spectral Sequential (Spe-Seq) Cell into a Spectral Graph Convolution module. In this section, we first introduce the motivation and architecture of the StemGNN block, and then briefly describe the Spe-Seq Cell and Spectral Graph Convolution module separately.

**StemGNN Block**   Spectral Graph Convolution has been widely used in time-series forecasting task due to its extraordinary capability of learning latent representations of multiple time-series in the spectral domain. The key component is applying Graph Fourier Transform (GFT) to capture inter-series relationships. It is worth noting that the output of GFT is also a multivariate time-series while GFT does not learn intra-series temporal relationships explicitly. Therefore, we can utilize Discrete Fourier Transform (DFT) to learn the representations of the input time-series on the trigonometric basis in the frequency domain, which captures the repeated patterns in the periodic data or the auto-correlation features among different timestamps. Motivated by this, we apply the Spe-Seq Cell on the output of GFT to learn temporal patterns in the frequency domain. Then the output of the Spe-Seq Cell is processed by the rest components of Spectral Graph Convolution.

Our model can also be extended to multiple channels. We apply GFT and Spe-Seq Cell on each individual channel $X_i$ of input data and sum the results after graph convolution with kernel $\Theta_{\cdot j}$. Next, Inverse Graph Fourier Transform (IGFT) is applied on the sum to obtain the $j$th channel $Z_j$ of the output, which can be written as follows,

$$Z_j = \mathcal{GF}^{-1}\left(\sum_i g_{\Theta ij}(\Lambda_i)\mathcal{S}(\mathcal{GF}(X_i))\right). \tag{4}$$

Here $\mathcal{GF}$, $\mathcal{GF}^{-1}$ and $\mathcal{S}$ denote GFT, IGFT, and Spe-Seq Cell respectively, $\Theta_{ij}$ is the graph convolution kernel corresponding to the $i$th input and the $j$th output channel, and $\Lambda_i$ is the eigenvalue matrix of normalized Laplacian and the number of eigenvectors used in GFT is equivalent to the multivariate dimension ($N$) without dimension reduction. After that we concatenate each output channel $Z_j$ to obtain the final result $Z$.

Following [19], we use learnable parameters to represent basis vectors $V$ and a fully-connected layer to generate basis expansion coefficients $\theta$ based on $Z$. Then the output can be calculated by a combination of different bases: $Y = V\theta$. We have two branches of this module in the StemGNN block, one to forecast future values, namely forecasting branch, and the other to reconstruct history values, namely backcasting branch (denoted by $B$). The backcasting branch helps regulate the functional space for the block to represent time-series data.

Furthermore, we employ residual connections to stack multiple StemGNN blocks to build deeper models. In our case, we use two StemGNN blocks. The second block tries to approximate the residual between the ground-truth values and the reconstructed values from the first block. Finally, the outputs from both blocks are concatenated and fed into GLU and fully-connected layers to generate predictions.

**Spectral Sequential Cell (Spe-Seq Cell)**   The Spe-Seq Cell $\mathcal{S}$ aims to decompose each individual time-series after GFT into frequency basis and learn feature representations on them. It consists of four components in order: Discrete Fourier Transform (DFT, $\mathcal{F}$), 1D convolution, GLU and Inverse Discrete Fourier Transform (IDFT, $\mathcal{F}^{-1}$), where DFT and IDFT transforms time-series data between temporal domain and frequency domain, while 1D convolution and GLU learn feature representations in the frequency domain. Specifically, the output of DFT has real part ($\hat{X}_u^r$) and imaginary part ($\hat{X}_u^i$), which are processed by the same operators with different parameters in parallel. The operations can be formulated as:

$$M^*(\hat{X}_u^*) = \text{GLU}(\theta_\tau^*(\hat{X}_u^*), \theta_\tau^*(\hat{X}_u^*)) = \theta_\tau^*(\hat{X}_u^*) \odot \sigma^*(\theta_\tau^*(\hat{X}_u^*)), * \in \{r, i\} \tag{5}$$

where $\theta_\tau^*$ is the convolution kernel with the size of 3 in our experiments, $\odot$ is the Hadamard product and nonlinear sigmoid gate $\sigma^*$ determines how much information in the current input is closely related to the sequential pattern. Finally, the result can be obtained by $M^r(\hat{x}_u^r) + iM^i(\hat{x}_u^i)$, and IDFT is applied on the final output.

**Spectral Graph Convolution**   The Spectral Graph Convolution [13] is composed of three steps. (1) The multivariate time-series input is projected to the spectral domain by GFT. (2) The spectral representation is filtered by a graph convolution operator with learnable kernels. (3) Inverse Graph Fourier Transform (IGFT) is applied on the spectral representation to generate final output.

*Graph Fourier Transform (GFT)* [8] is a basic operator for Spectral Graph Convolution. It projects the input graph to an orthonormal space where the bases are constructed by eigenvectors of the normalized graph Laplacian. The normalized graph Laplacian [1] can be computed as: $L = I_N - D^{-\frac{1}{2}}WD^{-\frac{1}{2}}$,

where $I_N \in \mathbb{R}^{N \times N}$ is the identity matrix and $D$ is the diagonal degree matrix with $D_{ii} = \sum_j W_{ij}$. Then, we perform eigenvalue decomposition on the Laplacian matrix, forming $L = U \Lambda U^T$, where $U \in \mathbb{R}^{N \times N}$ is the matrix of eigenvectors and $\Lambda$ is a diagonal matrix of eigenvalues. Given multivariate time-series $X \in \mathbb{R}^{N \times T}$, the operators of GFT and IGFT are defined as $\mathcal{GF}(X) = U^T X = \hat{X}$ and $\mathcal{GF}^{-1}(\hat{X}) = U\hat{X}$ respectively. The graph convolution operator is implemented as a function $g_\Theta(\Lambda)$ of eigenvalue matrix $\Lambda$ with parameter $\Theta$. The overall time complexity is $O(N^3)$

## 5 Experiments

### 5.1 Setup

Table 1: Summary of Datasets

|  | METR-LA | PEMS-BAY | PEMS07 | PEMS03 | PEMS04 | PEMS08 | Solar | Electricity | ECG5000 | COVID-19 |
|---|---|---|---|---|---|---|---|---|---|---|
| # of nodes | 207 | 325 | 228 | 358 | 307 | 170 | 137 | 321 | 140 | 25 |
| # of timesteps | 34,272 | 52,116 | 12,672 | 26,209 | 16,992 | 17,856 | 52,560 | 26,304 | 5,000 | 110 |
| Granularity | 5min | 5min | 5min | 5min | 5min | 5min | 10min | 1hour | - | 1day |
| Start time | 9/1/2018 | 1/1/2018 | 7/1/2016 | 5/1/2012 | 7/1/2017 | 3/1/2012 | 1/1/2006 | 1/1/2012 | - | 1/22/2020 |

We compare the performances of StemGNN on nine public datasets, ranging from traffic, energy and electrocardiogram domains with other state-of-the-art models, including FC-LSTM [26], SFM [32], N-BEATS [19], DCRNN [17], LSTNet [14], ST-GCN [31], DeepState [21], TCN [3], Graph Wavenet [29] and DeepGLO [25]. We tune the hyper-parameters on the validation data by grid search for StemGNN. Finally, the channel size of each graph convolution layer is set as 64 and the kernel size of 1D convolution is 3. Following [31], we adopt the RMSprop optimizer, and the number of training epochs is set as 50. The learning rate is initialized by 0.001 and decayed with rate 0.7 after every 5 epochs. We use the Mean Absolute Errors (MAE) [11], Mean Absolute Percentage Errors (MAPE) [11], and Root Mean Squared Errors (RMSE) [11] to measure the performances, which are averaged by $H$ steps in multi-step prediction. We report the performances of baseline models in their original publications unless otherwise stated. The dataset statistics are summarized in Table 1.

We conduct the dataset into three part for training, validation and testing with a ratio of 6:2:2 on PEMS03, PMES04, PEMS08, and 7:2:1 on META-LA, PEMS-BAY, PEMS07, Solar, Electricity and ECG. The inputs of ECG are normalized by min-max normalization following [5]. Besides, the inputs are normalized by Z-Score method [19]. That means StemGNN is trained on normalized input where each time-series in the training set is re-scaled as $X_{in} = (X_{in} - \mu(X_{in}))/\sigma(X_{in})$, where $\mu$ and $\sigma$ denote the mean and standard deviation respectively. More details descriptions about datasets, evaluation metrics, and experimental settings can be found in Appendix B, C and D.

### 5.2 Results

The evaluation results are summarized in Table 2, and more details can be found in Appendix E.1.Generally, StemGNN establishes a new state-of-the-art on most of the datasets. Furthermore, the model does not need apriori topology and demonstrates the feasibility of learning latent correlations automatically. In particular, on all datasets, StemGNN improves an average of 8.1% on MAE and 13.3% on RMSE compared to the best baseline for each dataset. In terms of baseline models, FC-LSTM only takes temporal information into consideration and performs estimation in the time domain. SFM models the time-series data in the frequency domain and shows stable improvement over FC-LSTM. Besides, N-BEATS, TCN and DeepState are state-of-the-art deep learning models specialized for sequential modeling. A common limitation is that they do not capture the correlations among multiple time-series explicitly, hindering their application to multivariate time-series forecasting. Therefore, it is natural that StemGNN shows much better performances against these baselines.

On the other hand, spatial and temporal correlations can be modeled in GNN-based approaches, such as DCRNN, ST-GCN and GraphWaveNet. However, most of them need a pre-defined topology of different time-series and are not applicable to Solar, Electricity and ECG datasets. GraphWaveNet is able to work without a pre-defined structure but the performance is not satisfied. For traffic forecasting tasks, StemGNN outperforms these models consistently without any prior knowledge of the road network. It is convincing that a data-driven latent correlation layer works more effectively than human defined priors. Moreover, DeepGLO is a hybrid method that enables the model to focus both on

Table 2: Forecasting results on different datasets

| | MAE | RMSE | MAPE(%) | MAE | RMSE | MAPE(%) | MAE | RMSE | MAPE(%) |
|---|---|---|---|---|---|---|---|---|---|
| | METR-LA [12] | | | PEMS-BAY [4] | | | PEMS07 [4] | | |
| FC-LSTM [26] | 3.44 | 6.3 | 9.6 | 2.05 | 4.19 | 4.8 | 3.57 | 6.2 | 8.6 |
| SFM [32] | 3.21 | 6.2 | 8.7 | 2.03 | 4.09 | 4.4 | 2.75 | 4.32 | 6.6 |
| N-BEATS [19] | 3.15 | 6.12 | 7.5 | 1.75 | 4.03 | 4.1 | 3.41 | 5.52 | 7.65 |
| DCRNN [17] | 2.77 | 5.38 | 7.3 | 1.38 | 2.95 | 2.9 | 2.25 | 4.04 | 5.30 |
| LSTNet [14] | 3.03 | 5.91 | 7.67 | 1.86 | 3.91 | 3.1 | 2.34 | 4.26 | 5.41 |
| ST-GCN [31] | 2.88 | 5.74 | 7.6 | 1.36 | 2.96 | 2.9 | 2.25 | 4.04 | 5.26 |
| TCN [3] | 2.74 | 5.68 | 6.54 | 1.45 | 3.01 | 3.03 | 3.25 | 5.51 | 6.7 |
| DeepState [21] | 2.72 | 5.24 | 6.8 | 1.88 | 3.04 | 2.8 | 3.95 | 6.49 | 7.9 |
| GraphWaveNet [29] | 2.69 | 5.15 | 6.9 | 1.3 | 2.74 | 2.7 | - | - | - |
| DeepGLO [25] | 2.91 | 5.48 | 6.75 | 1.39 | 2.91 | 3.01 | 3.01 | 5.25 | 6.2 |
| **StemGNN (ours)** | **2.56** | **5.06** | **6.46** | **1.23** | **2.48** | **2.63** | **2.14** | **4.01** | **5.01** |
| | PEMS03 [4] | | | PEMS04 [4] | | | PEMS08 [4] | | |
| FC-LSTM [26] | 21.33 | 35.11 | 23.33 | 27.14 | 41.59 | 18.2 | 22.2 | 34.06 | 14.2 |
| SFM [32] | 17.67 | 30.01 | 18.33 | 24.36 | 37.10 | 17.2 | 16.01 | 27.41 | 10.4 |
| N-BEATS [19] | 18.45 | 31.23 | 18.35 | 25.56 | 39.9 | 17.18 | 19.48 | 28.32 | 13.5 |
| DCRNN [17] | 18.18 | 30.31 | 18.91 | 24.7 | 38.12 | 17.12 | 17.86 | 27.83 | 11.45 |
| LSTNet [14] | 19.07 | 29.67 | 17.73 | 24.04 | 37.38 | 17.01 | 20.26 | 31.96 | 11.3 |
| ST-GCN [31] | 17.49 | 30.12 | 17.15 | 22.70 | 35.50 | 14.59 | 18.02 | 27.83 | 11.4 |
| TCN [3] | 18.23 | 25.04 | 19.44 | 26.31 | 36.11 | 15.62 | 15.93 | 25.69 | 16.5 |
| DeepState [21] | 15.59 | 20.21 | 18.69 | 26.5 | 33.0 | 15.4 | 19.34 | 27.18 | 16 |
| GraphWaveNet [29] | 19.85 | 32.94 | 19.31 | 26.85 | 39.7 | 17.29 | 19.13 | 28.16 | 12.68 |
| DeepGLO [25] | 17.25 | 23.25 | 19.27 | 25.45 | 35.9 | 12.2 | **15.12** | 25.22 | 13.2 |
| **StemGNN (ours)** | **14.32** | 21.64 | **16.24** | **20.24** | **32.15** | **10.03** | 15.83 | **24.93** | **9.26** |
| | Solar [14] | | | Electricity [2] | | | ECG [5] | | |
| FC-LSTM [26] | 0.13 | 0.19 | 27.01 | 0.62 | 0.2 | 24.39 | 0.32 | 0.54 | 31.0 |
| SFM [32] | 0.05 | 0.09 | 13.4 | 0.08 | 0.13 | 17.3 | 0.17 | 0.58 | 11.9 |
| N-BEATS [19] | 0.09 | 0.15 | 23.53 | - | - | - | 0.08 | 0.16 | 12.428 |
| LSTNet [14] | 0.07 | 0.19 | 19.13 | 0.06 | 0.07 | 14.97 | 0.08 | 0.12 | 12.74 |
| TCN [3] | 0.06 | 0.06 | 21.1 | 0.072 | 0.51 | 16.44 | 0.1 | 0.3 | 19.03 |
| DeepState [21] | 0.06 | 0.25 | 19.4 | 0.065 | 0.67 | 15.13 | 0.09 | 0.76 | 19.21 |
| GraphWaveNet [29] | 0.05 | 0.09 | 18.12 | 0.071 | 0.53 | 16.49 | 0.19 | 0.86 | 19.67 |
| DeepGLO [25] | 0.09 | 0.14 | 21.6 | 0.08 | 0.14 | 15.02 | 0.09 | 0.15 | 12.45 |
| **StemGNN (ours)** | **0.03** | **0.07** | 11.55 | **0.04** | **0.06** | 14.77 | **0.05** | **0.07** | **10.58** |

Table 3: Results for ablation study of the PEMS07 dataset

| | **StemGNN** | w/o LC | w/o Spe-Seq Cell | w/o DFT | w/o GFT | w/o Residual | w/o Backcasting |
|---|---|---|---|---|---|---|---|
| MAE | **2.144** | 2.158 | 2.612 | 2.299 | 2.237 | 2.256 | 2.203 |
| RMSE | **4.010** | 4.017 | 4.692 | 4.170 | 4.068 | 4.155 | 4.077 |
| MAPE(%) | **5.010** | 5.113 | 6.180 | 5.336 | 5.222 | 5.230 | 5.130 |

local properties of individual time-series as well as global properties, while multivariate correlations are encoded by a matrix factorization module. It shows competitive performances on some datasets like solar and PEMS08, but StemGNN is generally more advantageous. Arguably, it is beneficial to recognize both structural and sequential patterns jointly in the spectral domain.

## 5.3 Ablation Study

To better understand the effectiveness of different components in StemGNN, we design six variants of the model and conduct ablation study on several datasets. Table 3 summarizes the results obtained on PEMS07 [4], and more results on other datasets can be found in Appendix E.2.

The results show that all the components are indispensable. Specifically, **w/o Spe-Seq Cell** indicates the importance of temporal patterns for multivariate time-series forecasting. The Discrete Fourier Transform inside the cell also brings benefits as verified by **w/o DFT**. Furthermore, **w/o Residual** and **w/o Backcasting** demonstrate that both residual and backcasting designs can learn supplementary information and enhance time-series representation. **w/o GFT** shows the advantages of leveraging GFT to capture structural information in a graph. Moreover, we use a pre-defined topology instead of correlations learned by the *Latent Correlation Layer* in **w/o LC**, which indicates the superiority of StemGNN for learning inter-series correlations automatically.

# 6 Analysis

## 6.1 Traffic Forecasting

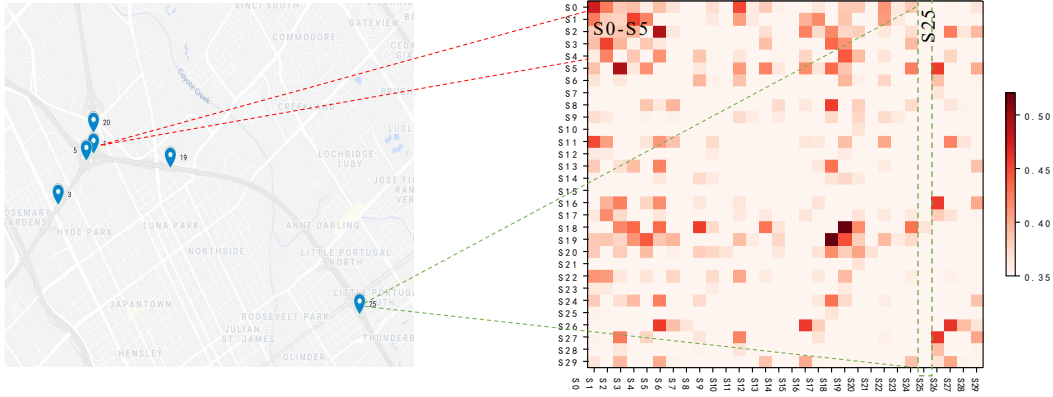

Figure 2: The adjacent matrix obtained from latent correlation layer.

To investigate the validity of our proposed latent correlation layer, we perform a case study in the traffic forecasting scenarios. We choose 6 detectors from PEMS-BAY and show the average correlation matrix learned from the training data (the right part in Figure 2). Each column represents a sensor in the real world. As shown in the figure, column $i$ represents the correlation strength between detector $i$ and other detectors. As we can see, some columns have a higher value like column $s_1$, and some have a smaller value like column $s_{25}$. This indicates that some nodes are closely related to each other while others are weakly related. This is reasonable, since detector $s_1$ is located near the intersection of main roads, while detector $s_{25}$ is located on a single road, as shown in the left part of Figure 2. Therefore, our model not only obtains an outstanding forecasting performance, but also shows an advantage of interpretability.

## 6.2 COVID-19

Table 4: Forecasting results (MAPE%) on COVID-19

|  | FC-LSTM [26] | SFM [32] | N-BEATS [19] | TCN [3] | DeepState [21] | GraphWaveNet [29] | DeelpGLO [25] | StemGNN (ours) |
|---|---|---|---|---|---|---|---|---|
| 7 Day | 20.3 | 19.6 | 16.5 | 18.7 | 17.3 | 18.9 | 17.1 | **15.5** |
| 14 Day | 22.9 | 21.3 | 18.5 | 23.1 | 20.4 | 24.4 | 18.9 | **17.1** |
| 28 Day | 27.4 | 22.7 | 20.4 | 26.1 | 24.5 | 25.2 | 23.1 | **19.3** |

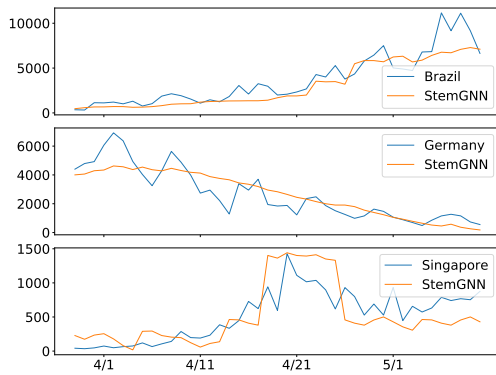

(a) Forecasting result for the 28th day

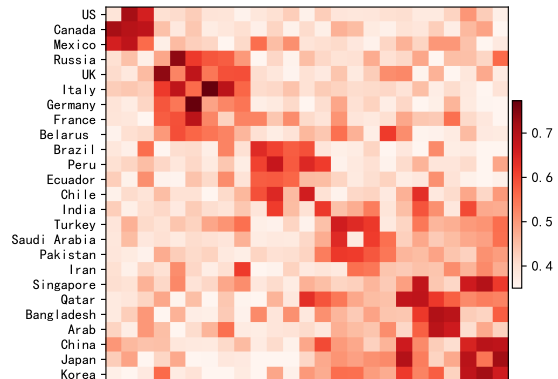

(b) Inter-country correlations

Figure 3: Analysis on COVID-19

To investigate the feasibility of StemGNN for real problems, we conduct additional analyses on daily number of newly confirmed COVID-19 cases. We select the time-series of 25 countries with severe COVID-19 outbreak from 1/22/2020 to 5/10/2020 (110 days). Specifically, we use the first 60 days for training and the rest 50 days for testing. In this analysis, we forecast the values of $H$ days in the future, where $H$ is set to be 7, 14 and 28 separately. Table 4 shows the evaluation results where we can see that StemGNN outperforms other state-of-the-art solutions in different horizons.

Figure 3(a) illustrates the forecasting results of Brazil, Germany and Singapore in advance of 28 days. Specifically, we set $H = 28$ and take the predicted value of the 28th day for visualization. Each timestamp is predicted with the historical data four weeks before that timestamp. As shown in the figure, the predicted value is consistent with the ground truth. Taking Singapore as an example, after 4/14/2020, the volume has rapidly increased. StemGNN forecasts such trend successfully in advance of four weeks.

The dependencies among different countries learned by the *Latent Correlation Layer* are visualized in Figure 3(b). Larger numbers indicate stronger correlations. We observe that the correlations captured by StemGNN model are in line with human intuition. Generally, countries adjacent to each other are highly correlated. For example, as expected, US, Canada and Mexico are highly correlated to each other, so are China, Japan and Korea.

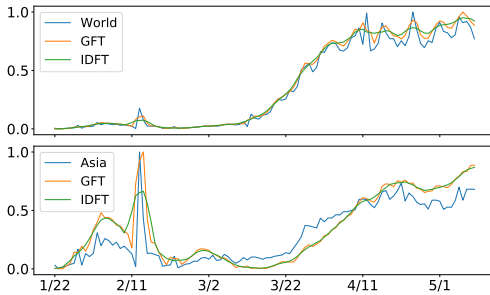

Figure 4: Effectiveness of GFT and DFT.

We further analyze the effect of GFT and DFT in StemGNN. We choose the top two eigenvectors obtained by eigenvalue decomposition of the normalized Laplacian matrix $L$ and visualize their corresponding time-series after GFT in Figure 4. As encoded by the eigenvectors, the first time-series captures a common trend in the world and the second time-series captures a common trend from Asian countries. For a clear comparison, we also visualize the ground truth of daily number of newly confirmed in the whole world and Asian countries. As shown in Figure 4, the time-series after GFT capture these two major trends obviously. Moreover, the time-series data in the spectral space becomes smoother, which increases the generalization capability and reduces the difficulty of forecasting. We also draw the time-series after processed by the *Spectral Sequential Cell* (denoted by IDFT in Figure 4), which recognizes the data patterns in a frequency domain. Compared to the ones after GFT, the result time-series turn to be smoother and more feasible for forecasting.

## 7 Conclusion

In this paper, we propose a novel deep learning model, namely Spectral Temporal Graph Neural Network (StemGNN), to take the advantages of both inter-series correlations and temporal dependencies by modeling them jointly in the spectral domain. StemGNN outperforms existing approaches consistently in a variety of multivariate time-series forecasting applications. Future works are considered in two directions. First, we will investigate approximation method to reduce the time complexity of StemGNN, because directly applying eigenvalue decomposition is prohibitive for very large graphs of high-dimensional time-series. Second, we will look for its application to more real-world scenarios, such as product demand, stock price prediction and budget analysis. We also plan to apply StemGNN for predictive maintenance, which is an important topic in AIOps.

## Broader Impact

Time-series analysis is an important research domain for machine learning, while multivariate time-series forecasting is one of the most prevalent tasks in this domain. This paper proposes a novel model, StemGNN, for the task of multivariate time-series forecasting. For the first time, we model the inter-series correlations and temporal patterns jointly in the spectral domain, which improves the representation power of multivariate time-series. Signals in the time domain can be easily restored by the orthogonal basis in the frequency domain, so we could leverage the rich information beneath the hood of the frequency domain to improve the forecasting results. StemGNN is neat yet powerful as proved by extensive experiments and analyses. It is one of the first attempts that incorporate Discrete Fourier Transform with Graph Neural Networks. We believe it will motivate more exploration along this direction in other related domains with temporal features, such as social graph mining and sentiment analysis. Moreover, StemGNN adopts a latent correlation layer in an end-to-end framework to learn relationships among multivariate signals automatically. This makes StemGNN a general approach that could be applied to a wide range of applications, including surveillance of traffic flows, healthcare data monitoring, natural disaster forecasting and economy.

Multivariate time-series forecasting has significant societal implications as well. A sophisticated supply chain management system may be built if we can predict market trend precisely. It also brings benefit to our daily life. For example, there is a real case about 'Flooding Risk Analysis'. The task is to predict when there will be a flooding in certain areas near the city. The prediction mainly depend on two external factors, tides and rainfalls. Accurate prediction can alert people to keep away from the area at the corresponding time to avoid unnecessary losses. For COVID-19, accurate prediction of the trend may help the government make suitable decisions to control the spread of the epidemic. According to a case study on COVID-19 in this paper, we can reasonably forecast the daily number of newly confirmed cases four weeks in advance based on historical data. Nevertheless, how to predict the trend from the beginning without sufficient historical data is more challenging and remained to be investigated. Moreover, we are aware of the negative impact of this technique to infringement of personal privacy. Customers' behavior may be predicted by unscrupulous business persons on historical records, which provides a convenient way to send spam information. Hackers may also use the predicted data to avoid surveillance of a bank's security system for fraud credit card transactions.

Although current models are still far away from predicting future data absolutely correct, we do believe that the margin is decreasing rapidly. We hope that researchers could understand and mitigate the potential risks in this domain. We would like to mention the concept of responsible AI, which guides us to integrate fairness, interpretability, privacy, security, accountability into the design of AI systems. We suggest researchers to take a people-centered approach to research, development, and deployment of AI and cultivate a responsible AI-ready culture.

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
