[Supplementary Material]

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

# A   Notation

Table 5: Notations

| | |
|---|---|
| $\mathcal{G}$ | multivariate temporal graph |
| $X$ | multivariate time-series input, $X_t \in \mathbb{R}^N$ is observed values at timestamp $t$ |
| $\Delta_\theta$ | all parameters in the network |
| $W$ | adjacency matrix, where $w_{ij} \in W$ indicates the strength of edge $ij$ |
| $N$ | the number of time-series |
| $K$ | the number of previous time steps |
| $H$ | the number of future time steps to forecast (horizon) |
| $B$ | the entire network that generates backcasting output |
| $\hat{X}$ | forecasted time-series output, $\hat{X}_t \in \mathbb{R}^N$ is the value at timestamp $t$ |
| $R$ | the last hidden state of attention mechanism |
| $Q, K$ | query and key in the attention mechanism |
| $W^Q, W^K$ | learnable parameters for query and key projections |
| $\Theta_{\cdot j}$ | graph convolution kernel |
| $\mathcal{GF}$ | Graph Fourier Transform |
| $\mathcal{GF}^{-1}$ | Inverse Graph Fourier Transform |
| $\mathcal{S}$ | the Spe-Seq Cell |
| $V$ | basis vectors |
| $Z$ | the output after IGFT |
| $Y$ | the forecasting output |
| $\mathcal{F}$ | Discrete Fourier Transform |
| $\hat{X}_u^*$ | the real part $\hat{X}_u^r$ and imaginary part $\hat{X}_u^i$ after DFT |
| $\mathcal{F}^{-1}$ | Inverse Discrete Fourier Transform |
| $\theta_\tau^*$ | the convolution kernel of Spe-Seq Cell |
| $L$ | the normalized graph Laplacian |
| $U$ | the matrix of eigenvectors |
| $\Lambda$ | the diagonal matrix of eigenvalue |

# B   Reproduction details for StemGNN

## B.1   Datasets

We compare the performance of StemGNN with other state-of-the-art models on ten public datasets, ranging from traffic, energy, electrocardiogram to COVID-19 domain. Among all the datasets, only the datasets from traffic domain provide apriori topology. Table 1 shows the statistics of these datasets.

**Traffic Forecasting.** These datasets are collected by the Caltrans Performance Measurement System (PeMS) [4] and the loop detectors in the highway of Los Angeles County (METR) [13]. The monitoring data is aggregated by 5 minutes from 30-second data samples, which means there are 12 points in the flow data for each hour. We evaluate the performance of traffic flow forecasting on PEMS03, PEMS07, PEMS08 and traffic speed forecasting on PEMS04, PEMS-BAY and METR-LA.

**Energy Forecasting.** We consider two datasets in this perspective. (1) Solar. It contains photo-voltaic production of 137 stations in Alabama State [15], which is sampled every 10 minutes. (2) Electricity. It contains hourly time-series of electricity consumption from 370 customers [2].

**Electrocardiogram Forecasting.** We adopt the ECG5000 dataset from the UCR time-series Classification Archive [5], and this dataset is composed of 140 electrocardiograms (ECG) with a length of 5000.

**COVID-19 Trend Forecasting.** This dataset is provided by the Center for Systems Science and Engineering (CSSE) at Johns Hopkins University[3], which contains daily case reports including confirmed, deaths and recovered number. We use the daily number of newly confirmed COVID-19

cases as the time-series and select the time-series of 25 countries with severe COVID-19 outbreak from 1/22/2020 to 5/10/2020 (totally 110 days). Specifically, we use the first 60 days for training and the rest 50 days for testing.

## B.2 Metrics

Let $\hat{X}_t$ and $X_t$ be the predicted and ground truth values at timestamp $t$ respectively, T is the total number of timestamps. The evaluation metrics we use in the experiments can be computed by:

$$MAE = \frac{1}{T} \sum_{t=1}^{T} |X_t - \hat{X}_t|, \tag{6}$$

$$MAPE = \frac{1}{T} \sum_{t=1}^{T} \left| \frac{X_t - \hat{X}_t}{X_t} \right| \times 100\%, \tag{7}$$

$$RMSE = \sqrt{\frac{1}{T} \sum_{t=1}^{T} (X_t - \hat{X}_t)^2}. \tag{8}$$

## C Reproduction details for baselines

FC-LSTM [29]: FC-LSTM can forecast univariate time-series with fully-connected LSTM hidden units. The source code can be found at `https://github.com/farizrahman4u/seq2seq`. We use 4 stacked LSTM cells of 1000 hidden size and other detailed settings can be referred to [29].

SMF [36]: SMF improves the LSTM model to be able to break down the cell states of a given univariate time-series into a series of different frequency components. We use SMF by setting hidden dimension as 50, frequence dimension as 10. Other default configurations are given in the source code: `https://github.com/z331565360/State-Frequency-Memory-stock-prediction`.

N-BEATS [21]: N-BEATS proposes a deep neural architecture based on backward and forward residual links and a very deep stack of fully-connected layers without using time-series domain knowledge. We use the open source code from: `https://github.com/philipperemy/n-beats`, and only modify the data I/O interface for different shapes of inputs. In our experiments, the backcast length is 10 and the hidden units number is 128. According to the recommendation, we turn on the 'share_weights_in_stack' option.

LSTNet [15]: takes advantage of the convolution layer to discover the local dependence patterns among multi-dimensional input variables, and the recurrent layer to captures the complex long-term dependency patterns. We use the open source code from: `https://github.com/fbadine/LSTNet`, and modify the data shapes of inputs. In our experiments, the number of output filters in the CNN layer is 100 and the CNN filter size is 6. Other experimental settings we refer to papers and code default values.

DCRNN [18]: DCRNN is a deep learning framework for traffic forecasting that incorporates both spatial and temporal dependencies in the traffic flow. Some of the results of DCRNN are directly reported in [18, 28], and we use the source code at `https://github.com/liyaguang/DCRNN` when reproduction is necessary. The horizon size is 12 and the RNN layer number is 2 with 64 units in our experiments. DCRNN is not applicable in scenarios without a priori topology. Thus, DCRNN is only used to forecast the traffic data.

STGCN [35]: STGCN is a novel deep learning framework for traffic prediction, integrating graph convolution and gated temporal convolution through spatio-temporal convolutional blocks. The performances of STGCN can be found at [35] and the source code is available at `https://github.com/VeritasYin/STGCN_IJCAI-18`. We use 12 history steps to forecast future data with batch size as 50, epoch number as 50, and learning rate as 0.001. STGCN is not applicable to scenarios without a priori topology.

TCN [3]: TCN combines best practices such as dilations and residual connections with the causal convolutions for autoregressive prediction. We take the source code at `https://github.com/locuslab/TCN`. We use a configuration similar to polyphonic music task mentioned in this paper,

where the kernel size is 5, the gradient clip is 0.2, the upper epoch limit is 100 and the initial learning rate is 0.001.

DeepState [23]: This model marries state space models with deep recurrent neural networks. First, it uses recurrent neural networks to calculate $h_t = RNN(h_{t-1}, x_t)$. Then, this model uses $h_t$ to calculate the parameters of state space $\Theta_t = \Phi(h_t)$ . Finally, the likelihood $pss(z_{1:T}|\Theta_{1:T})$ are calculated and the parameters are learned through maximum log likelihood. DeepState is integrated in Gluon Time Series (GluonTS), which is the Gluon toolkit for probabilistic time series modeling. The tutorials can be found at `https://gluon-ts.mxnet.io/`. We use the default configuration given by the tool and only change its sampling frequency as given in Table 1.

GraphWaveNnet [32]: GraphWaveNet is a method that represents each node's network neighborhood via a low-dimensional embedding by leveraging heat wavelet diffusion patterns. The results of Graph Wavenet are reported at [32, 34, 28], and the source code can be found at `https://github.com/nnzhan/Graph-WaveNet`. We turn on the 'add graph convolution layer' option when reproduce on some datasets. We set the weight decay rate as 0.0001 and dropout rate as 0.3. Other configurations follow the options recommended in the paper. We use the priori topology when it is available (traffic forecasting), and this method also works in scenarios without a priori topology (energy forecasting, electrocardiogram forecasting and COVID-19 forecasting).

DeepGLO [27] : This model leverages both global and local features during training and forecasting. The global component, TCN regularized Matrix Factorization (TCN-MF), captures global patterns by representing each of the original time-series as a linear combination of $k$ basis time-series, and we set $k = 128$ in our experiments. We use the default setting of DeepGLO provided by `https://github.com/rajatsen91/deepglo`. It has two batch sizes: horizontal batch size (set to 256) and vertical batch size (set to 128). Besides, we change the start time and frequency of different datasets. The kernel size is set as 7 for both hybrid model and local model, and the learning rate is set to be 0.005. We report the best results from the normalized and unnormalized settings in the paper.

Please refer to their publications for more detailed descriptions and settings.

# D    Experiment Details

We conduct all our experiments using one NVIDIA GeForce GTX 1080 GPU. We divide the dataset into three part for training, validation and testing according to [10] (PEMS03, PMES04, PEMS08), [35] (PEMS07), and [18] (META-LA, PEMS-BAY, Solar, Electricity, ECG). The inputs of ECG are normalized by min-max normalization following [5]. Besides, the inputs are normalized by Z-Score method [21]. That means StemGNN is trained on normalized input where each time-series in the training set is re-scaled as $X_{in} = (X_{in} - \mu(X_{in}))/\sigma(X_{in})$, where $\mu$ and $\sigma$ denote the mean and standard deviation respectively. The evaluation of Solar, Electricity and ECG datasets is performed on the re-scaled data following [5] and [23], i.e., first using the normalization algorithm to transform Solar, Electricity and ECG into a value range of $[0, 1]$, and then applying StemGNN to generate the forecasting values. Afterwards, the predictions are transformed back to the original scale, and the metrics are calculated on the original data.

In StemGNN, the dimension of self-attention layer is 32, which is chosen from a search space of [16, 32, 64, 128] on the validation data. The channel size of each graph convolution layer is 64 chosen from a search space of [16, 32, 64, 128] and the kernel size of 1D convolution is 3, selected from a search space of [3, 6, 9, 12]. The batch size is 50. The learning rate is initialized as 0.001 and decays with rate 0.7 after every 5 epochs. The total number of training epochs is set as 50.

In the traffic datasets (METR-LA, PEMS-BAY, PEMS07, PEMS07, PEMS04, PEMS08), the data is aggregated by 5 minutes, so the number of timestamps per day is 288. For traffic speed forecasting task, we use the one-hour historical data to predict the next 15 minutes data [18, 35]; for traffic flow forecasting task, we use one-hour historical data to predict the values in the next hour [10]. The solar dataset is aggregated every 10 minutes, according to [23, 27], we forecast the trend in future 0.5 hour with 4-hour historical data. For the electricity data, we follow [23, 27] which use 24-hour historical data to infer the values in next 3 hours. For ECG5000 dataset, according to [21], we set the forecasting step as 3 and the sliding window size as 12. For COVID-19 dataset, we forecast the future 4 weeks' trend, which means the max forecasting step is 28 (days). Besides, we use averaging MAE, MAPE, RMSE over the predicted time period to evaluate StemGNN and all baselines.

# E  Results

## E.1  More results on METR-LA and COVID-19

Table 6: Forecasting Results on METR-LA and COVID-19

| | | MAE | RMSE | MAPE(%) | MAE | RMSE | MAPE(%) | MAE | RMSE | MAPE(%) |
|---|---|---|---|---|---|---|---|---|---|---|
| | | | 15min | | | 30min | | | 1hour | |
| | FC-LSTM [29] | 3.44 | 6.3 | 9.60 | 3.77 | 7.23 | 10.90 | 4.37 | 8.69 | 13.20 |
| | SFM [36] | 3.21 | 6.2 | 8.7 | 3.37 | 6.68 | 9.62 | 3.47 | 7.61 | 10.15 |
| | N-BEATS [21] | 3.15 | 6.12 | 7.5 | 3.62 | 7.01 | 9.12 | 4.12 | 8.04 | 11.5 |
| | DCRNN [18] | 2.77 | 5.38 | 7.30 | 3.15 | 6.45 | 8.80 | 3.6 | 7.59 | 10.50 |
| METR-LA [13] | STGCN [35] | 2.88 | 5.74 | 7.60 | 3.47 | 7.24 | 9.60 | 4.59 | 9.4 | 12.70 |
| | TCN [3] | 2.74 | 5.68 | 6.54 | - | - | - | - | - | - |
| | DeepState [23] | 2.72 | 5.24 | 6.8 | 3.13 | 6.16 | 8.31 | 3.61 | 7.42 | 10.8 |
| | GraphWaveNet [32] | 2.69 | 5.15 | 6.90 | 3.07 | 6.22 | 8.40 | 3.53 | 7.37 | 10 |
| | DeepGLO [27] | 2.91 | 5.48 | 6.75 | 3.36 | 6.42 | 8.33 | 3.66 | 7.39 | 10.3 |
| | **StemGNN (ours)** | **2.56** | **5.063** | **6.46** | **3.011** | **6.03** | **8.23** | **3.43** | **7.23** | **9.85** |
| | | | 7Day | | | 14Day | | | 28Day | |
| | FC-LSTM [29] | 1803.65 | 3284.77 | 20.3 | 2135.54 | 3855.75 | 22.9 | 2554.07 | 4318.4 | 27.4 |
| | SFM [36] | 1699.85 | 3499.25 | 19.6 | 1812.82 | 3589 | 21.3 | 1851 | 3720 | 22.7 |
| | N-BEATS [21] | 594.43 | 928.37 | 16.5 | 847.14 | 1286.36 | 18.5 | 882.42 | 1349.46 | 20.4 |
| COVID-19 [7] | TCN [3] | 662.24 | 2363.95 | 18.7 | 1307 | 2871.17 | 23.1 | 2117.34 | 3419.3 | 26.1 |
| | DeepState [23] | 922.87 | 1982.32 | 17.3 | 1852.73 | 2091.32 | 20.4 | 2345.3 | 2386.4 | 24.5 |
| | GraphWaveNet [32] | 1056.1 | 1227.3 | 18.9 | 1899.5 | 2125.7 | 24.4 | 2331.5 | 2451.9 | 25.2 |
| | DeepGLO [27] | 1131.23 | 1023.19 | 17.1 | 1718.69 | 1734.67 | 18.9 | 2084.51 | 2291.19 | 23.1 |
| | **StemGNN (ours)** | **462.24** | **718.11** | **15.5** | **533.67** | **871.17** | **17.1** | **662.24** | **1023.19** | **19.3** |

In order to prove that there is a steady improvement for multi-step forecasting, we choose METR-LA and COVID-19 for an evaluation of longer time span. As shown in Table 6, StemGNN achieves excellent performance in multi-steps forecasting scenarios. In particular, we use COVID-19 data to forecast the number of infected people in the next 1-4 weeks which is of great significance to help relevant departments make decisions. Compared to other solutions, StemGNN reduces the time-dependent error accumulation and improves the performance of long-term forecasting.

## E.2  More results for ablation study

Table 7: Ablation results

| | | MAE | RMSE | MAPE(%) | MAE | RMSE | MAPE(%) | MAE | RMSE | MAPE(%) |
|---|---|---|---|---|---|---|---|---|---|---|
| | | | 15min | | | 30min | | | 45min | |
| | **StemGNN** | **2.144** | **4.01** | **5.01** | **2.994** | **5.35** | **7.25** | **3.158** | **6.34** | **8.43** |
| | w/o Latent Correlations | 2.158 | 4.017 | 5.113 | 3.004 | 5.525 | 7.303 | 3.214 | 6.496 | 8.672 |
| | w/o Spe-Seq Cell | 2.612 | 4.692 | 6.189 | 3.459 | 6.257 | 8.448 | 4.505 | 8.241 | 11.343 |
| PEMS07 [4] | w/o DFT | 2.299 | 4.17 | 5.336 | 3.183 | 5.945 | 7.532 | 3.817 | 7.145 | 9.058 |
| | w/o GFT | 2.237 | 4.068 | 5.222 | 3.065 | 5.755 | 7.355 | 3.691 | 6.922 | 8.899 |
| | w/o Residual | 2.256 | 4.155 | 5.23 | 3.073 | 5.854 | 7.357 | 3.684 | 7.021 | 8.918 |
| | w/o Backcasting | 2.203 | 4.077 | 5.13 | 3.034 | 5.641 | 7.316 | 3.394 | 6.912 | 8.681 |
| | | | 15min | | | 30min | | | 1hour | |
| | **StemGNN** | **2.56** | **5.063** | **6.46** | **3.011** | **6.03** | **8.23** | **3.43** | **7.23** | **9.85** |
| | w/o Latent Correlations | 2.79 | 5.24 | 6.867 | 3.122 | 6.922 | 8.36 | 3.568 | 7.462 | 9.97 |
| | w/o Spe-Seq Cell | 3.077 | 5.71 | 6.99 | 3.491 | 7.072 | 8.83 | 3.905 | 7.906 | 10.163 |
| METR-LA [13] | w/o DFT | 2.81 | 5.37 | 6.93 | 3.24 | 6.95 | 8.52 | 3.717 | 7.571 | 9.99 |
| | w/o GFT | 2.867 | 5.25 | 6.891 | 3.201 | 6.92 | 8.41 | 3.701 | 7.552 | 10.02 |
| | w/o Residual | 2.83 | 5.29 | 6.71 | 3.228 | 6.57 | 8.27 | 3.724 | 7.471 | 9.95 |
| | w/o Backcasting | 2.85 | 5.219 | 6.57 | 3.06 | 6.233 | 8.51 | 3.56 | 7.72 | 10.03 |

- **w/o Latent Correlations (LCs).**   We use a priori topology instead of automatic correlations. As shown in Table 7, dynamic latent correlations performs even better than a static priori topology. The reason may be that a priori topology is static, but the StemGNN is capable of building a topology for each sliding window dynamically, which captures the newest knowledge about the interaction between different time-series.

- **w/o Spe-Seq Cell.**   This setting does not equip with the *Spe-Seq Cell*. It performs the worst among all settings, indicating that temporal dependency is the most important clue for time-series forecasting.

- **w/o DFT.**   It removes DFT and inverse DFT operators in the Spectral Sequential Cell. Thus, temporal dependencies are modeled in the time domain. It shows improvement over the naive baseline without *Spe-Seq Cell*, but under-performs StemGNN by a large margin, which proves the benefit of DFT.

- **w/o GFT.**   We no longer use the entire StemGNN cell, but only take the Spectral Sequential Cell. The performance drops significantly, which shows a necessity of capturing latent correlations through graph Fourier transform.

- **w/o Residual.**   This setting has two stacked StemGNN blocks without residual connection. It verifies that the second block learns supplement information through a residual connection.

- **w/o Backcasting.**   This model disables the backcasting branch, showing the benefit of backcasting module for enhancing time-series representation.

# F   Analysis

## F.1   Efficiency Analysis

Table 8: Results of efficiency analysis

|  | Training time (in seconds) | | | |
| --- | --- | --- | --- | --- |
|  | StemGNN | SFM | N-BEATS | STGCN |
| PEMS07 [4] | 459 | 1013 | 1251 | **352** |
| METR-LA [13] | 1137 | 2284 | 2561 | **1035** |

Although the time complexity is $O(N^3)$ w.r.t. the multivariate dimension $N$, training process on all the datasets can be finished in a reasonable time. For a concrete comparison, we summarize the training time of StemGNN, SFM, N-BEATS and STGCN on the PEMS07 and METR-LA datasets separately. The results are shown at Table 8. StemGNN is similar to the first-order approximate graph convolution model (STGCN) in time, but our performance is improved significantly. Comparing to other baselines, StemGNN has a superior training speed, and inference speed shows the same conclusion.

## F.2   Learning Curve Comparison

Figure 5: Learning curves on METR-LA.

The learning curves for StemGNN and major baselines on METR-LA dataset are shown in Figure 5, where the x-axis denotes wall-clock time and the y-axis denotes validation RMSE. It shows that StemGNN has an effective training procedure and achieves a better RMSE score at convergence than other SOTAs with comparable training times.

## F.3   Case Study on COVID-19

To investigate the usability and interpretability of StemGNN, we conduct a detailed analysis on COVID-19 data. We assume the daily number of newly confirmed cases as time-series, and choose 25 countries with severe outbreak as multivariate input. Figure 6(a) shows the visualization of the inter-series correlations captured automatically by our model. In this figure, row $i$ represents the correlation strength between country $i$ and other countries. As we can see, correlations are not uniform across countries. This indicates that some nodes are closely related to neighboring countries while weakly related to others. This is reasonable since countries on the same continent have higher correlations in population mobility and related policies. Therefore, our model not only obtains the best forecasting performance, but also shows the advantage of interpretability.

To prove that the conversion of graph Fourier transform over multivariate time-series data is effective, we first visualize the matrix of eigenvectors ($U$) obtained by the decomposition of the normalized Laplace matrix $L$ on Figure 6(c). Each column of $U$ represents an eigenvector corresponding to a eigenvalue sorted from the highest to the lowest. We select three eigenvectors with the largest eigenvalues ($u_0 - u_2$) and visualize the corresponding time-series after GFT for further analysis. As shown in Figure 6(c), $u_0$ captures the general trend countries across the world, $u_1$ learns the major trend of Asian countries, and $u_2$ learns the common trend of South American countries. As illustrated in Figure 6(b), the three components capture these three trends respectively, and the time-series in the spectral space is relatively smooth [19] compared to the original data, reducing the difficulty of forecasting. Thus, it is clear that graph Fourier transform can better leverage the relationships

(a) Visualization of adjacent matrix

(b) Real world data and graph Fourier time-series

(c) Visualization of eigenvector

(d) Forecasting results on COVID-19

Figure 6: The latent correlations and forecasting results on COVID-19

learned by the latent correlation layer and make the forecasting easier through feature smoothness. Moreover, IDFT also helps to improve the smoothness of time-series and lead to better generalization (Figure 6(b)). Finally, Figure 6(d) shows the forecasting results for several exemplar countries and demonstrate the feasibility of StemGNN. In the figure, 'Canada' represents for ground truth; 'Canada1' means forecasting in advance of 1 day; 'Canada7' means forecasting in advance of 7 days. It is similar for other countries.

Figure 6 and Figure 7 show the result comparison of stemGNN and two major baselines which respectively use historical data to forecast the number of confirmed people in advance of one day (denoted by *-1) and one week (denoted by *-7). We select several typical countries, and the features show that StemGNN can predict the future trends more accurately. Thanks to graph Fourier transform and Spectral Sequential Cell, StemGNN captures the major trends more smoothly and predict the changes of data more timely. The turning points predicted by other baselines have larger time delays compared to StemGNN.

Figure 7: The latent correlations and forecasting results on COVID-19