[Reviews · NeurIPS 2020]

Review 1

Summary and Contributions: This paper pays attention to multivariate time-series forecasting problem. The author proposes StemGNN which incorporates inter-series correlations and intra-series patterns by GFT and DFT respectively. In detail, the authors define a new structure which is called stemGNN block to solve both the learning pattern work and prediction work. Furthermore, StemGNN is made up of two stemGNN block which make full use of the data and the residual. To authors’ knowledge, this paper is the first one to adopt such ideas.

Strengths: 1. This paper combines pattern extraction from inter-series and intra-series together which offers a new way to make predictions in multivariate time series. Tradition methods of predicting multivariate time series depends on a prior multivariate dependency which is always defined manually, the authors use GFT to model inter-series correlation. Furthermore, DFT is used to capture temporal dependencies in intra-series which is jointly finished with GFT in spectral domain. 2. Structure of StemGNN is made up of two StemBlock defined by the author and they make full use of both data and its residual which is helpful to learn the features of intra-series. Besides, it uses both forecast and backcast results to train which are stated elaborately in the paper. The pattern and relationship can be depicted clearly by this model. 3. Experiments on 9 real-word datesets show good result on three metrics: MAE, RMSE, MAPE and show great generality as well. Besides, the authors also design several ablation study to prove the significance of each part.

Weaknesses: 1. Stacked network does bring more powerful computing competence but on the other hand, the more complex the network is, the more its training costs. It would be better if you can show the variation of metrics during training process. 2. In test phase, how long does the model cost to make predictions. If it has a time complexity of linear or sublinear to the length of predictions, applications will be easier. 3. In the w/o residual ablation experiment, residual connection is removed instead of deleting the second stemGNN block. I wonder how much the performance will decline if the second block is removed.

Correctness: This paper proposes a brand new idea to solve multivariate time-series prediction problem by incorporating intra patterns and inter relations. Relations are revealed by latent correlation layer and patterns are learned by an seq-to-seq cell. We can treat it right on the basis of experiment results.

Clarity: In general, this paper is well written. However, in line 190 of page 5: “ is the convolution kernel with the size of 3 in our experiments, is the the Hadamard product and nonlinear sigmoid gate determines how much information in the current input is closely related to the sequential pattern”, a redundant word “the” exists in this sentence. Another place which can be done better is that a notion table is missing. Several characters are used in the paper and several variants of them are also used to represent different states. When I was reading this paper, for some times I need to find and check the meaning of some ordinary character.

Relation to Prior Work: In the second chapter, the author discusses a lot on related works and analyses the weakness and differences from their proposed model. First and foremost, the author divides time series forecasting into two directions: univariate technique and multivariate technique. Then it states several univariate techniques, such as FC-LSTM, N-BEATS and analyses the weakness of this kind of technique which can be summarized as without considering correlations between different time series. About multivariate techniques, the authors list a lot of examples. Traditional methods such as TCN and DeepGLO tends to treat multivariate time series as a tensor and use matrix techniques to learn the features. DCRNN, ST-GCN, GraphWaveNet combine graph convolutional network into the network to capture dependencies such as spatio-temporal dependencies. However, the biggest weakness of them is they either ignore the inter-series correlations or require a dependency graph as a prior. Besides, to the author’s knowledge, the proposed StemGNN is the first to combine the task of capturing temporal patterns and multivariate dependencies jointly in the spectral domain. These two points give a big difference from previous contributions.

Reproducibility: Yes

Additional Feedback: Figure 2 in analysis on COVID-19 shows that this model can only predict the trend and fluctuations (especially in Germany subfigure) are missed a lot. Maybe this is caused by data quality and it is better to add more prediction figure on other datasets. Stacked network is being used more broadly, however, the more complex the network is, the huger computation cost it requires. I wonder how big the difference is between single StemBlock and two StemBlocks. Furthermore, I am wondering the variation of metrics during training process because I think sometimes the dataset cannot support a successful training of a deep network. As you mentioned at the last of paper, “directly applying eigenvalue decomposition is prohibitive for very large graphs high-dimensional time-series”. I also hope information about the test phase: how long does the model cost to make predictions. I believe a time complexity of linear or sublinear to the length of predictions may help the applications be easier.


Review 2

Summary and Contributions: The authors propose a novel neural network architecture for forecasting multivariate time series. The method estimates a correlation matrix, which along with the data is fed into a StemGNN layer. The StemGNN layer is boosted once. The whole network is trained to minimize a joint forecasting and backcasting loss. The StemGNN transforms the data and correlation matrix into the spectral domain to linearly decouple the components. It then featurizes the components and transforms them back to the time domain, yielding both a forecast

Strengths: The authors present a strong set of comparisons and ablative evidence in favor of their claims that the propose architecture establishes a new state-of-the-art in multivariate time-series forecasting. The method is conceptually appropriate and appears sound.

Weaknesses: The method appears to assume a static correlation matrix. How would the method perform if correlations where not static? The authors make statements suggesting the method identifies "clear patterns" (ln 13) but give only a correlation matrix as an example. I find the automatic calculation of series correlations to be only a small benefit. One could after all compute the empirical covariance matrix from data directly. To be sure, the network can tune its believed covariances to enhance predictive performance, so joint calculation is different than direct calculation.

Correctness: The methods and evaluation appear correct.

Clarity: The paper is well written and easy to follow. Figure 2 could use a more expressive caption.

Relation to Prior Work: Prior work and its connection to the presented material is clearly discussed.

Reproducibility: Yes

Additional Feedback: In the rebuttal the authors have clarified my concerns. I would ask that they try to make the role of the correlation matrix as clear as possible in the camera ready version.


Review 3

Summary and Contributions: This paper addresses the problem of multivariate time-series prediction. The premise of the problem is, given N possibly correlated time series, predict the next H time steps for each of the time series. The paper develops over existing methods by proposing a novel deep neural network based algorithm that simultaneously accounts for the “spatial” and temporal correlations. The proposed algorithm first constructs an adjacency matrix to capture the similarity between the different time series by using a self-attention based similarity measure. Post this, the data is passed through two “stemGNN” blocks, with each block as described below. The data X (of size NxT, where N is the number of nodes and T the number of time steps) is then transformed into the eigenspace of the above adjacency matrix. The transformed set of N time series are then passed through a DFT block, and then through a 1D convolutional layer. The signal is now is the spectral domain along both the spatial and temporal axes. A graph convolution operation is then applied using the explicit eigen decomposition computed earlier. Finally, each the time series are transformed back into the canonical domain and passed through two separate neural networks, one for forecasting each series and the other for “backcasting”. Contributions: The main contribution of the papers are: 1. A novel approach to handling multivariate time-series prediction problem by first applying a spectral transformation on both the spatial and temporal axes. 2. A self-attention based graph adjacency matrix construction method that can help identify connections across different time-series 3. Improved empirical performance across 9 different public datasets and demonstration of the utility of the model on COVID-19 cases prediction. %%%% Edit after rebuttal %%%%%% The authors have sufficiently addressed my concerns. I already had a good recommendation for this paper and will retain it. %%%%%%%%%%%%%%%%%%

Strengths: Strengths: 1. The paper is very clearly written and is easy to understand. The proposed method is thoroughly validated using experiments, including ablation studies on the model and a high-level analysis of the model’s predictions on the COVID-19 dataset. I especially appreciate the visualization of the eigenvectors. 2. The proposed method has a good empirical performance on the datasets considered and does better than all of the baselines considered. The gain in performance is considerable (if not stellar). However, the proposed idea is interesting and justifies the good performance. 3. The proposed architecture is well-motivated and the idea of using operations in the 2D spectral domain is very interesting.

Weaknesses: Weaknesses: 1. The proposed model has many good attributes, but a disadvantage that sort of stand out to me is the complexity of explicitly computing the eigen-decomposition of the graph Laplacian. In my opinion, the merits of the paper outweigh this drawback, but it is something the paper can improve upon. The O(N^3) complexity can easily be prohibitive on large datasets. For example, the COVID analysis is done at a country level. If this resolution were to be increased to state/county level, the resulting graph would be massive. I wonder if using approximations of the eigen-modes affect the performance of the model. Also, going by the results of the ablation study, removing the GFT leads to only an incremental reduction in performance. So I wonder how critical the GFT is, given that it is an expensive operation. 2. The paper can also mention/ give examples where the time-series prediction did not do great. For example, in the COVID-19 dataset, UK and Russia are learnt to be highly correlated, even though are not neighboring countries. Further, were there countries where the predictions were not great? How did the trajectories of these countries relate to the top eigenvectors of the Laplacian?

Correctness: The proposed method and the experiments are technically sound.

Clarity: The paper is very clear to read.

Relation to Prior Work: Mostly yes. Although after a search for relevant papers, I noticed that citation [10] is never actually referred to in the paper (I double-checked this, but there is a small chance I missed it). I wonder how the results of this paper compare against the proposed method. In general, it may not be a good idea to include a paper only in the references and not mention it in the body of the paper.

Reproducibility: Yes

Additional Feedback: I really enjoyed reading this paper and I liked the proposed ideas.


Review 4

Summary and Contributions: The paper proposes an architecture for multivariate time-series forecasting that manages to capture inter and intra series correlations in the frequency domain without requiring a prior knowledge domain. the architecture is experimentally validated through an ablation study that indicates that all of its portions are indispensable.

Strengths: the model outperforms existing models on 3 benchmarks and a real world study while being able to identify plausible correlations among the time-series.

Weaknesses: 1) the authors do not compare with the model in [15]: "Modeling long-and short-term temporal patterns with deep neural networks.". This restricts the potential impact of the model. 2) the model has many components whose hyper parameters are not fully provided (someone may have to trace them in the source code) 3) the paper doesn't propose a conceptual/computational novelty. it combines existing modules to achieves its results.

Correctness: 1) the authors should report learning curves to demonstrate convergence of the learning algorithm 2) for the experiments in table 1, how much is H?

Clarity: the paper was incoherent at some points. In particular, 1) in figure 1, it is not explained what are the Yi-s 2) in equation (2), B() (the backcasting lost) is not defined. 3) in equation (3) are w^q, w^k learnable parameters? 4) what is the number of the eigenvectors used in the gft??

Relation to Prior Work: the major difference, that is clearly described, from previous works is that the proposed model uses a data-driven approach for identifying inter-series correlations

Reproducibility: No

Additional Feedback: I would suggest the order of the descriptions in section 4.3 match the order of the components of the model in figure 1. for example, the description of Spectral Graph Convolution shouldn't be placed before the Spe-Seq Cell)

[Author Response · NeurIPS 2020]

**To All Reviewers:** We would like to thank all reviewers for your time and insightful feedback. Our full version (18 pages with appendix) already contains the answers to some of your questions, which can be found in the supplementary zip file in our original submission. We address your major concerns briefly here, and will incorporate all comments carefully in the revised version.

Figure 1: Learning curve comparison on METR-LA.

A common concern is about the time complexity and training/inference costs. A fixed sliding window is used for long sequences, so the time complexity is linear to the length of input data. Although the time complexity is $O(N^3)$ w.r.t. the multivariate dimension $N$, training process on all the datasets can be finished within a reasonable time. As shown in Table 7 (P15, L570), the training time of StemGNN is acceptable comparing with other SOTAs. For one epoch, StemGNN uses 459s on PEMS07 and 1137s on METR-LA, while the fastest SOTA uses 352s on PEMS07 and 1035s on METR-LA. We can see that StemGNN has acceptable efficiency while achieving better accuracy. When $N$ is extremely large, we can derive more practical solutions by dimension reduction techniques (e.g., randomized SVD), while the experimental study is left to future work. The learning curve on METR-LA dataset is shown in Figure 1, where the x-axis denotes wall-clock time and the y-axis denotes validation RMSE. It shows that StemGNN has an effective training procedure and achieves a superior RMSE at convergence.

**To Reviewer 1:** We will fix the typos in the text and include a notation table in the Appendix. More prediction figures on COVID-19 datasets are available in Figure 6 (Appendix E.3 (P18)). As shown in the figures, StemGNN is able to make reasonable predictions for the trend. However, fluctuations are much more difficult to be estimated, which may be caused by random noises. We will add more prediction figures on other datasets in the revised version.

After deleting the second StemGNN block, the 15min/30min/45min MAE drops by 5.4%/3.9%/15% on PEMS07, and the 15min/30min/1hour MAE drops by 8.8%/11.9%/11.5% on METR-LA, which indicates the effectiveness of the second StemGNN block.

**To Reviewer 2:** StemGNN generates a dynamic correlation matrix for each sliding window by the Latent Correlation Layer. We visualize the average correlation matrix from all sliding windows for clearer insights.The dynamic evolution will be analyzed in a future research. Moreover, capturing dynamic correlations jointly in the model is beneficial. You may refer to the ablation results in Table 2 (P7, L233) and Table 6 (Appendix D.2 (P14, L548)), where the "w/o LC" setting utilizes a static covariance matrix from SOTA. Especially for the METR-LA dataset (Table 6), we can see a significant performance drop (8% drop on MAE), showing the advantage of the Latent Correlation Layer.

**To Reviewer 3:** Removing GFT leads to a significant reduction in performance, as shown in "w/o GFT" of Table 6 (Appendix D.2 (P14, L548)). Especially on the METR-LA dataset, there is a 10% drop on MAE. In Figure 5(c) (Appendix E.3 (P16, L591)), we show the trajectories of countries related to the top eigenvectors. It is clear that the top three eigenvectors correspond to countries in the world, Asia, and South America, respectively. In addition, citation [10] has been moved to Appendix C due to space limitation. We will make the revised version self-contained.

**To Reviewer 4:** First, we'd like to emphasize our novelty and contributions. (1) The designed model for multivariate time-series forecasting is non-trivial, and many endeavors explore along this direction. Our work is the first that *models both inter-series and intra-series correlations jointly in the spectral domain* and achieves significant performance gains than the best SOTA for various applications (13.3% average RMSE surge on 9 datasets). (2) The model can be generalized on all multivariate time-series without predefined topologies, which solves the pain point of existing SOTAs [16, 33]. (3) As shown in Appendix E.2 & E.3 (P15-18), the learned attention matrices are explainable by humans, which benefits the transparency and credibility of the model.

We have reproduced the LSTNet [15] on all datasets. As a result, StemGNN outperforms LSTNet by 17%/21%/14% for MAE/RMSE/MAPE on the average of nine datasets. Therefore, it does not restrict the potential impact of our model. We have a detailed description of hyper-parameters for StemGNN in Appendix C (P13, L513). For example, the values of $H$ on all datasets can be seen in lines 529-539 of Appendix C. Appendix B (P12, L458) introduces the settings to reproduce SOTAs. We will include a notation table for better presentation in the revised version.

In addition, we give brief answers for clarity here and will polish the corresponding parts in the final version. (1) $Y_i$ in Figure 1 means the $i^{th}$ output channel of Y. (2) In equation 2, $B$ indicates the entire network that generates backcasting outputs (refer to L175). (3) $w^q$, $w^k$ are learnable parameters in the attention mechanism. (4) The number of eigenvectors used in GFT is equivalent to the multivariate dimension ($N$) without dimension reduction.

[Meta-Review · NeurIPS 2020]

Four expert reviewers have recommended acceptance of the paper and I agree with them. However, the final version of the paper should reflect a request from Reviewer #4 to include specific empirical comparisons.